# HBDrug3D: A Dataset and Benchmark for AI-Driven Heterobifunctional Molecule Design

## Abstract

As a vital branch of fragment-based drug design (FBDD), linker design generates molecular bridges that fuse two fragments into a complete compound and has gained widespread interest in AI-driven discovery with heterobifunctional modalities such as PROTACs, ADCs and PDCs. Linkers for these platforms must be custom-engineered to each biological mechanism, demanding geometric precision and physicochemical profiles far beyond those of conventional small-molecule linkers. However, existing PROTAC, ADC and PDC datasets remain nonstandardized, lack high-quality conformations and rely on inconsistent evaluation protocols and metrics, hampering robust model development. To address these gaps, we introduce HBDrug3D, the first benchmark dataset for heterobifunctional drug linker design. Firstly, we aggregated and stringently filtered raw data from three sources. Secondly, we harmonized storage formats and usage conventions across both chemical and engineering domains to establish a unified data-representation space. We then generated low-energy conformations with the OPLS4 force field and filtered out redundant or invalid structures. Finally, we leveraged our programmatic evaluation pipeline to survey diverse metrics, define HBDrug3D-specific criteria and benchmark state-of-the-art models. Results and case studies demonstrate that, while current methods can produce valid heterobifunctional linkers, substantial gains remain in overall performance and cross-modality robustness. Finally, we release an open-source codebase covering data preprocessing, model training, sampling and evaluation to lower adoption barriers and spur further research.

## 1 Introduction

Fragment-based drug design (FBDD) paradigms have garnered considerable attention in recent years (Xu & Kang, 2025). Especially, approaches integrating graph neural networks (GNNs) with deep generative models have dramatically accelerated the discovery of small molecules, antibodies, and peptides (Luong & Singh, 2023; Zhang et al., 2024; Aguilar Rangel et al., 2022; Li et al., 2024b). Studies show that in certain domains, these platforms now close the loop from in silico design to experimentally active molecules: generative models can sample biomolecules that are chemically valid, synthetically tractable and conformationally precise, and some are even ready for immediate preclinical evaluation (Yang et al., 2021; Luo et al., 2022; Mao et al., 2023). This progress rests on three pillars: GGNs' ability to encode intricate three dimensional (3D) molecular interactions; the diverse, tunable exploration enabled by generative architectures; and, critically, the abundance of high quality 3D structural data.

Heterobifunctional drugs, harnessing a dual-module synergistic mechanism and outstanding therapeutic benefits, have ushered in a revolutionary advance in cancer treatment (Békés et al., 2022; Tsuchikama et al., 2024; Dai et al., 2024). A prototypical heterobifunctional drug is assembled by covalently linking a tumor-targeting binding module to an effector module via a precise linker (Abeje et al., 2024; Ashman et al., 2022). This design enables highly specific recognition and targeting of cancer cells, suppressing their proliferation and metastasis to achieve potent antitumor efficacy while markedly reducing off-target toxicity (Bemis et al., 2021). As the critical bridge between modules, the linker not only governs the drug's conformational stability and pharmacokinetics but also dictates its targeting specificity, activity, and safety profile (Manzari et al., 2021). Representative platforms include Proteolysis Targeting Chimeras (PROTACs), Antibody-Drug Conjugates (ADCs), and Peptide-Drug Conjugates (PDCs): PROTACs recruit the E3 ligase system to selectively degrade

oncogenic proteins, whereas ADCs and PDCs exploit the targeting capabilities of antibodies or peptides, respectively, to deliver highly potent cytotoxins directly to tumor cells (Békés et al., 2022; Dumontet et al., 2023; Tsuchikama et al., 2024; Dai et al., 2024).

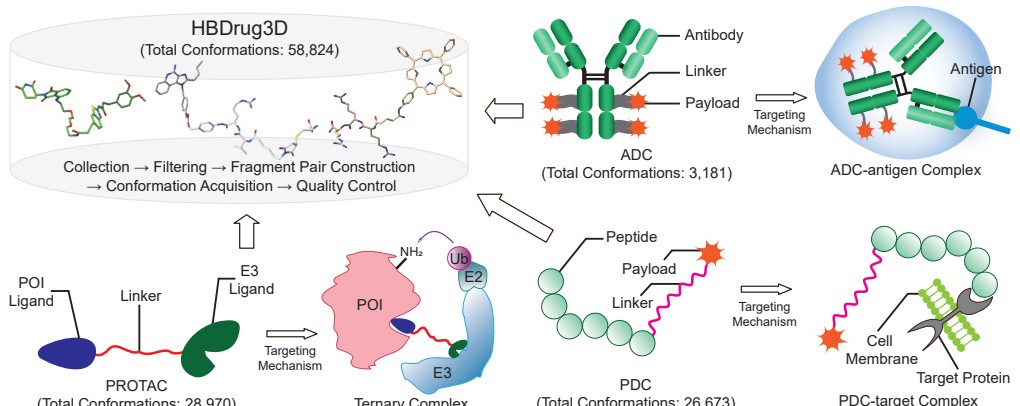

**Figure 1:** Overview of the HBDrug3D dataset. It comprises 59,314 conformations of PROTACs, ADCs and PDCs and displays representative structures such as the PROTAC ternary complex, the ADC-antigen complex and the PDC-target complex, illustrating the distinct targeting mechanisms of these heterobifunctional therapeutics.

To tackle these challenges, we present **HBDrug3D**, a comprehensive dataset for heterobifunctional drug linker design, as illustrated in Figure 1. It contains 6,279 heterobifunctional molecules and 59,314 conformations, spanning three druggable mechanisms. We also provide a unified codebase integrating data preprocessing, model sampling, evaluation, and state-of-the-art methods, all built upon a standardized benchmarking framework. **For dataset construction**, we treated conjugation sites in ADCs and peptide moieties in PDCs as parts of generalized compounds, enabling a unified representation across PROTACs, ADCs, and PDCs. Molecules were decomposed into functional fragments and linkers, with atomic-level mapping to define connectivity. Exhaustive 3D conformational sampling was performed using Schrödinger's ConfGen (Friesner et al., 2004), guided by the optimized potentials for liquid simulations (OPLS4) force field (Lu et al., 2021). Redundant or invalid conformers were removed based on root mean square deviation (RMSD) filtering to ensure structural diversity. **For benchmarking and evaluation**, we standardized experimental settings across all subsets and defined five metrics covering: **(i)** basic molecular properties, **(ii)** standardized 2D structural filtering, and **(iii)** 3D similarity assessment. These metrics were used to systematically assess the performance of baselines on HBDrug3D. To summarize, our main contributions are:

- Analysis of druggable mechanisms and linker size distributions reveals that heterobifunctional drug linker design is significantly more complex than small-molecule linker design, motivating its formal definition as a standalone task.

- Enabled by a unified molecular representation across PROTACs, ADCs, and PDCs, we built a high-quality 3D conformational dataset from the ground up to support heterobifunctional drug linker design.

- We established standardized experimental settings and evaluation metrics for heterobifunctional drug linker design, and systematically benchmarked the latest state-of-the-art models on the HBDrug3D dataset.

- The benchmarking results highlight the urgent need for more capable and generalizable linker design models based on 3D structures, in order to bridge the gap between small-molecule and heterobifunctional drug design.

## 2 RELATED WORK AND TAXONOMY

**Small Molecule Datasets.** These datasets are all related to small-molecule linker design, each tailored to a specific application scenario. The ZINC dataset (~439,000 examples) (Imrie et al., 2020a) involves

computationally generated 3D conformers and focuses on linking two fragments. The CASF dataset (309 examples) (Imrie et al., 2020a) provides experimentally determined 3D structures for fragment linking. The GEOM dataset (∼285,000 examples) (Igashov et al., 2024) supports more complex cases with three or more fragments and multiple linkers. The Pockets dataset (∼186,000 examples) (Igashov et al., 2024) incorporates protein binding site information to enable structure-conditioned linker generation. Figure 2 illustrates a substantial difference in linker size distribution between the ZINC dataset and hetero-bifunctional molecules. Figure 2 shows the distribution of linker sizes in the ZINC dataset, which differs significantly from the distribution observed in heterobifunctional molecules (See Figure 3).

**Heterobifunctional Molecule Datasets.** PROTAC-DB (Ge et al., 2025), ADCdb Shen et al. (2024), and PDCdb (Sun et al., 2025) are specialized online databases focusing on different classes of targeted therapeutics. PROTAC-DB includes detailed data on 6,111 PROTACs, 569 warheads, 2,753 linkers, and 107 E3 ligase ligands, with pharmacokinetic data and some ternary complex structures. ADCdb provides information on 6,572 ADCs, including their pharmaceutical and biological activities, with data from clinical trials, animal models, and cell lines. PDCdb focuses on 2,036 PDCs, detailing their biological activities, pharmaceutical properties, and chemical modifications. Table 1 compares all linker design-related datasets, including HBDrug3D, from three key aspects: whether the molecules are heterobifunctional, the availability of sufficient molecular conformations, and the diversity of those conformations.

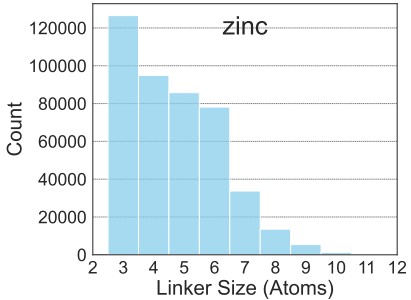

**Figure 2:** The linker size distributions in the zinc dataset. Small-molecule linkers span 3–12 atoms (mean ∼5; median 4)

**Table 1:** Datasets related to linker design tasks.

| Dataset | Het. Mol. | Conf. | Conf. Div. |
|---|---|---|---|
| ZINC | ✗ | ✓ | ✗ |
| CASF | ✗ | ✓ | ✗ |
| GEOM | ✗ | ✓ | ✗ |
| Pockets | ✗ | ✓ | ✗ |
| PROTAC-DB | ✓ | ✗ | ✗ |
| ADCdb | ✓ | ✗ | ✗ |
| PDCdb | ✓ | ✗ | ✗ |
| **HBDrug3D** | ✓ | ✓ | ✓ |

Heterobifunctional Molecule (Het. Mol.); Conformation (Conf.); Conformational Diversity (Conf. Div.)

**Linker-Based Drug Design.** Currently, six prominent methods have been developed for linker design: PROTAC-ZL (Zheng et al., 2022), DeLinker (Imrie et al., 2020a), 3DLinker (Huang et al., 2022), DiffLinker (Igashov et al., 2024), LinkerNet (Guan et al., 2023), and DiffPROTACs (Li et al., 2024a). PROTAC-ZL employs an augmented transformer with memory-assisted reinforcement learning to directly generate linker SMILES, enabling the design of PROTACs with desired properties. DeLinker and 3DLinker use variational frameworks for linker graph generation. DeLinker incorporates fragment distances and orientations, while 3DLinker improves the process through anchor atom prediction and simultaneous generation of linker graphs and 3D structures. Diffusion-based approaches such as DiffLinker and LinkerNet preserve geometric equivariance. DiffLinker automatically determines the number of linker atoms and supports multiple fragments, while LinkerNet removes the need to predefine fragment positions. DiffPROTACs introduces the O(3) Equivariant Graph Transformer (OEGT) for noise learning and 3D coordinate generation. Among these methods, PROTAC-ZL, LinkerNet, and DiffPROTACs have been exclusively applied to PROTAC design, while the remaining three have not yet been used for heterobifunctional drug design.

## 3   TASK AND DATASET

**Heterobifunctional Drug Linker Design** is a crucial component in the development of precision therapeutics, particularly in the design of PROTACs, ADCs, and PDCs. It involves creating linkers that effectively connect the functional fragments to the warhead or targeting moiety, ensuring optimal drug efficacy and specificity. In heterobifunctional drug design, we define linkers based on the following principles: **(i)** A linker must connect the warhead or targeting moiety to the functional fragment. **(ii)** The linker should possess sufficient flexibility to maintain the correct spatial orientation

and distance between the two components. **(iii)** The linker design should align with the pharmacological mechanisms specific to each class of heterobifunctional drugs. We represent both fragments and linkers as geometric graphs. Specifically, the fragment and linker are denoted as $\mathcal{G}^F = (\boldsymbol{v}^F, \boldsymbol{x}^F)$ and $\mathcal{G}^L = (\boldsymbol{v}^L, \boldsymbol{x}^L)$, respectively, where $\boldsymbol{v}^F$ and $\boldsymbol{v}^L$ are the sets of atomic numbers of the atoms in the fragment and linker, and $\boldsymbol{x}^F$ and $\boldsymbol{x}^L$ represent their corresponding atomic positions in 3D space. **Task Definition.** Given two molecular fragments $(\mathcal{G}^{F_1}, \mathcal{G}^{F_2})$, the objective is to train a generative model $p_\theta$ that learns the distribution of the linker $\mathcal{G}^L$. The model aims to generate linkers according to the conditional distribution $p_\theta(\mathcal{G}^L \mid \mathcal{G}^{F_1}, \mathcal{G}^{F_2})$, such that the generated linker connects the fragments into a conformationally stable and chemically valid molecule.

**HBDrug3D** dataset encompasses three categories of heterobifunctional molecules. Each entry in the dataset includes comprehensive molecular information, such as the SMILES, conformations, chemical bonds, and anchor atoms. The PROTAC subset contains 5,607 molecules, yielding a total of 28,970 conformations. It features 808 unique warheads and 184 E3 ligase ligands, connected via 2,379 distinct linkers. The linker lengths range from 5 to 43 atoms, with a mean length of ~14 and a median of 12. The ADC subset includes 246 molecules, generating 3,671 conformations. This subset comprises 17 conjugation sites and 26 payloads, bridged by 29 distinct linkers. The linker lengths span from 11 to 125 atoms, with an average of ~39 and a median of 37. The PDC subset consists of 426 molecules and 26,673 corresponding conformations. It contains 426 peptides and 114 payload types, assembled through 107 unique linkers. The linker lengths vary between 5 and 68 atoms, with an average of ~11 and a median of 6. Statistical details are summarized in Table 3 and Figure 3. Our data is avaliable at `https://anonymous.4open.science/r/HBDrug3D`.

**Table 2:** Statistics of the HBDrug3D dataset. For each subset, report counts and proportions of molecules and conformations, with a detailed breakdown of molecular composition.

| Subset | Molecules | | Conformations | | Warheads/ Conjugation Sites | Linkers | E3 Ligands/ Payloads |
|--------|-------|---------|-------|---------|-------|-------|-------|
| | Count | % total | Count | % total | | | |
| PROTAC | 5,607 | 89.3% | 28,970 | 48.8% | 808 | 2,379 | 184 |
| ADC | 246 | 3.9% | 3,671 | 6.2% | 17 | 29 | 26 |
| PDC | 426 | 6.8% | 26,673 | 45.0% | 426 | 107 | 114 |

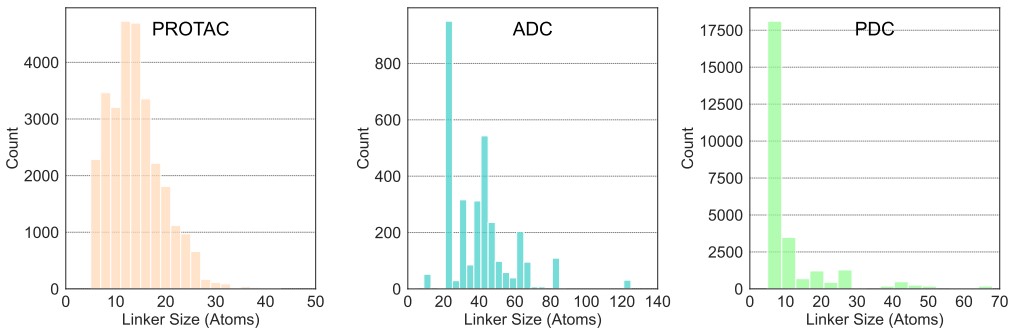

**Figure 3:** The linker size distributions in three subsets. PROTAC linkers span 5–43 atoms (mean ~14; median 12), ADC linkers span 11–125 atoms (mean ~39; median 37), and PDC linkers span 5–68 atoms (mean ~11; median 6).

## 4 DATA GENERATION

### 4.1 DATA COLLECTION AND FILTERING

The PROTAC, ADC, and PDC datasets were respectively obtained from three publicly available databases: PROTAC-DB (Ge et al., 2025), ADCdb (Shen et al., 2024), and PDCdb (Sun et al., 2025). Each PROTAC record contains the full molecular SMILES along with separate SMILES for its functional fragments, including the warhead and E3 ligase ligand, as well as the linker connecting

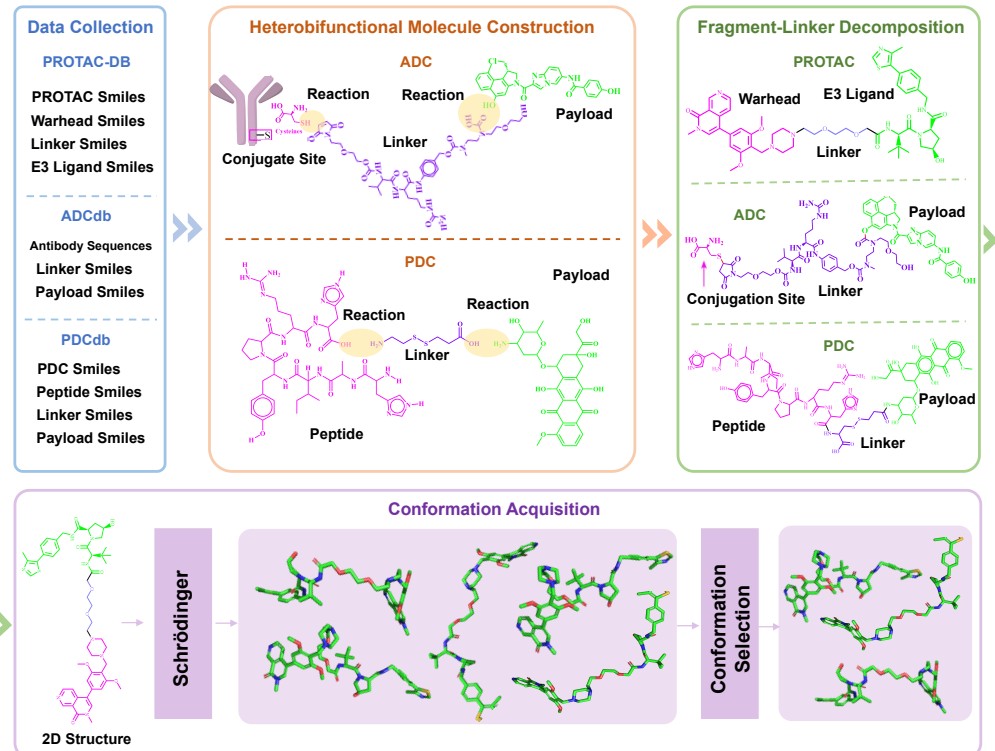

**Figure 4:** Overview of data generation process for the HBDrug3D dataset. Obtaining the structure of heterobifunctional molecules involves four sequential steps including data collection, heterobifunctional molecule construction, fragment-linker decomposition, and acquisition of conformations.

them. The ADC dataset provides detailed antibody conjugation sites, accompanied by SMILES for both the linker and payload components. For PDC molecules, each entry includes the full molecular SMILES and distinct SMILES for the peptide, payload, and the connecting linker. All datasets underwent rigorous quality control procedures, including the removal of **(i)** entries with missing or null values, **(ii)** records containing annotation errors in functional fragments or linkers, and **(iii)** molecules with non-physiological linker lengths less than 5 atoms. After this stringent curation, the final high-quality datasets comprise 5,607 PROTACs, 1,338 ADCs, and 426 PDCs.

### 4.2 FRAGMENT-LINKER DECOMPOSITION AND ATOMIC MAPPING

**PROTAC.** For the PROTAC dataset, we employed an atomic index-based segmentation strategy to annotate functional fragments and linkers. Specifically, the warhead, E3 ligase ligand, and linker were matched against the full molecular structure through substructure matching, allowing precise assignment of atomic indices to each component.

**ADC.** The ADC dataset annotation followed a standardized three-step workflow. Beginning with 1,338 initial entries, we first systematically classified antibody-linker conjugation sites, identifying 17 distinct patterns (complete catalog in Appendix A) and determining their corresponding conjugation mechanisms (documented in Appendix B). For each ADC, we then performed molecular modeling using ChemDraw (Li et al., 2004) to simulate two sequential reactions: **(1)** site-specific antibody-linker conjugation and **(2)** payload attachment to the linker, thereby constructing well-defined conjugation-linker-payload (CLP) molecules. Figure 5 illustrates this process using a trastuzumab-based ADC as a representative example, where interchain disulfide reduction generated reactive cysteine residues enabling selective conjugation. The workflow comprised three key stages: cysteine site identification, linker modification guided by reaction schemes, and controlled payload attachment. Application of this protocol across all entries yielded 246 CLPs. SMILES Sequences were extracted for all molecular components (conjugation sites, linkers, payloads), with atomic mapping achieved through substructure matching, consistent with the methodology applied to the PROTAC dataset.

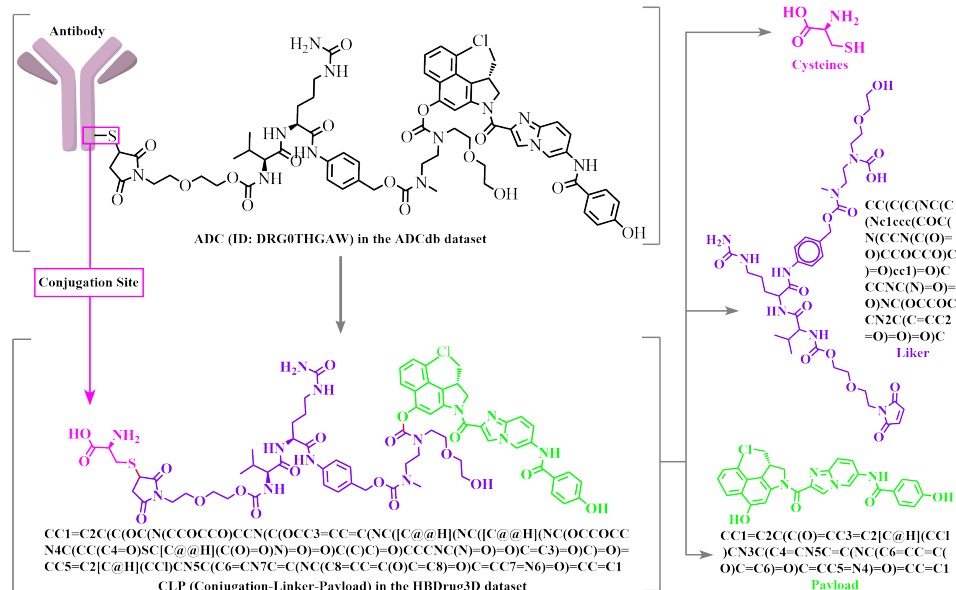

**Figure 5:** The extraction process of CLP in the HBDrug3D dataset. Taking the aforementioned ADC as an example, this workflow comprises three critical stages: cysteine site identification, reaction scheme-guided linker modification, and controlled payload attachment.

**PDC.** For PDCs, the dataset exclusively provided pre-reaction SMILES strings of individual components (peptide, linker, and payload). Due to the absence of direct atom-mappable structures, we established a manual annotation protocol. First, conjugation reactions were computationally simulated in ChemDraw (Li et al., 2004) to generate modified linker structures. These refined linkers then served as reference templates to systematically partition each PDC into three defined structural components: modified peptide, modified linker, and modified payload. SMILES representations were subsequently generated for each component, followed by atomic mapping using the standardized PROTAC annotation methodology.

## 4.3 Conformation Acquisition

All heterobifunctional molecules (PROTACs, ADCs, and PDCs) underwent exhaustive 3D conformational sampling using Schrödinger's ConfGen module (Friesner et al., 2004) with standardized parameters: an energy window of 10 kcal/mol above the global minimum, a maximum of 300 conformations per molecule, and an implicit solvent model (GBSA) at physiological pH $7.0 \pm 2.0$. Each conformation was energy-minimized with the OPLS4 force field (Lu et al., 2021) (heavy atom RMSD convergence threshold = 0.01 Å), followed by pairwise RMSD-based redundancy filtering (cutoff = 0.5Å) to ensure conformational diversity. This protocol generated 59,314 total conformations, comprising 28,970 from 5,607 PROTACs (5.2 conformations/molecule), 3,671 from 246 CLPs(12.9 conformations/CLP), and 26,673 from 426 PDCs (62.6 conformations/molecule). Notably, PDCs exhibited significantly higher conformational diversity than PROTACs or CLPs, a phenomenon attributed to the intrinsic flexibility of their peptide backbones. This high level of structural variability is especially relevant for modeling peptide-based linkers, which often adopt dynamic and extended geometries in solution.

## 5 Benchmarks

**Data Split.** We adopt a scaffold-based strategy to split the dataset. For each heterobifunctional molecule, its scaffold is extracted, and conformations sharing the same scaffold are assigned to the same subset. For the validation and test sets, we randomly select scaffolds and allocate their corresponding conformations into the sets until the target data size is reached. The number of scaffolds and conformations in each subset is summarized in Table 3.

**Table 3:** Data Split of PROTAC, PDC and ADC subsets.

| Subset | Train | | Valid | | Test | |
|--------|-------------|------------------|-------------|------------------|-------------|------------------|
| | # Scaffolds | # Conformations | # Scaffolds | # Conformations | # Scaffolds | # Conformations |
| PROTAC | 858 | 28,170 | 15 | 400 | 19 | 400 |
| ADC | 67 | 3,471 | 11 | 100 | 9 | 100 |
| PDC | 197 | 25,873 | 7 | 400 | 7 | 400 |

**Baseline Models.** We benchmark four state-of-the-art linker design models on our HBDrug3D dataset: (1) **DeLinker**(Imrie et al., 2020a) is a 2D distance-aware graph generative model that encodes 3D spatial relationships as edge attributes in graph neural networks; (2) **3DLinker**(Huang et al., 2022) is the first E(3)-equivariant variational autoencoder for end-to-end 3D linker generation with explicit anchor prediction; (3) **DiffLinker**(Igashov et al., 2024) is a SE(3)-equivariant diffusion model that implements a conditional denoising process for geometrically-aware linker generation; (4) **LinkerNet**(Guan et al., 2023) is a fragment-pose-aware diffusion model that incorporates rigid body dynamics constraints during the generative process.

## 5.1 Experimental Setup and Evaluation

**Experimental Setup.** We conducted comprehensive benchmarking of all four models (DeLinker, 3DLinker, DiffLinker, and LinkerNet) across three subsets of HBDrug3D: PROTAC (28,970 samples), ADC (3,671 samples), and PDC (26,673 samples). Each subset was partitioned into training/validation/test sets as mentioned above: PROTACs (28,170/400/400), ADCs (3,471/100/100), and PDCs (25,873/400/400). All experiments were performed on NVIDIA A100 GPUs. Detailed implementation specifications, including architecture configurations and training protocols for each model, are documented in Appendix C.

**Evaluation Metrics.** We evaluate our generated molecules from three complementary perspectives: (i) common generative metrics widely used in molecular design, including **validity** (chemical soundness), **novelty** (non-overlap with training samples), **uniqueness** (fraction of unique molecules), and **recovery rate** (exact matches with reference structures); (ii) chemical property-based metrics, such as passing the standardized **2D filtering** protocol (Imrie et al., 2020b) (incorporating ring aromaticity verification (Ertl & Schuffenhauer, 2009) and PAINS substructure screening (Baell & Holloway, 2010)) and synthetic accessibility (SA) score; (iii) 3D structural similarity metrics, specifically the **Shape and Color RDKit ( $SC_{RDKit}$) score** (Imrie et al., 2020a), which quantifies molecular resemblance by combining two complementary components: a Gaussian volume overlap assessing geometric alignment of conformers, and a color overlap capturing spatial correspondence of pharmacophoric features (hydrogen-bond donors/acceptors, aromatic rings, hydrophobic moieties, and charged groups). The score ranges from 0 to 1, with higher values indicating stronger 3D shape and chemical-feature similarity, thus reflecting how well the generative model reproduces both overall geometry and key pharmacophoric characteristics of the reference compounds.

## 5.2 Experimental Results and Analysis

**Model Performance.** Table 4 summarizes the evaluation results of baseline models on the PROTAC, ADC, and PDC subsets. Notably, the publicly available DeLinker and 3DLinker models failed to generate valid linkers for large molecular graphs in our HBDrug3D dataset. Therefore, our analysis mainly focuses on comparing LinkerNet and DiffLinker, evaluated across four core metrics: validity, novelty, uniqueness, and recovery rate.

(i) In terms of **validity**, both models achieve their best performance on the PROTAC subset, with the proportion of valid molecules ranging from 60% to 80%. The performance decreases on the ADC subset, and drops further on the PDC subset, where the validity rate is only around 20%.

(ii) For **novelty**, both models reach nearly 100% across all subsets, indicating that the generated molecules are structurally distinct from the training data and ensure sufficient diversity from existing samples.

(iii) Regarding **uniqueness**, both models maintain a high level, demonstrating their ability to generate a large number of non-duplicated molecules and thus avoid redundancy within the generated results.

**Table 4:** Evaluation results on three subsets in the HBDrug3D dataset. For each fragment pair on the test set, 100 candidates are generated for evaluation. Experiments were run for 3 times.

| Subset | Method | Validity, % | Novelty, % | Uniqueness, % | Recovery, % |
|--------|--------|-------------|------------|---------------|-------------|
| PROTAC | LinkerNet | 60.78 ± 0.29 | 100.00 ± 0.00 | 81.43 ± 4.67 | 35.62 ± 3.21 |
|        | DiffLinker | 79.75 ± 0.31 | 100.00 ± 0.00 | 79.18 ± 9.67 | 41.79 ± 9.23 |
| ADC    | LinkerNet | 30.98 ± 0.81 | 100.00 ± 0.00 | 98.65 ± 6.23 | 10.00 ± 1.58 |
|        | DiffLinker | 32.10 ± 0.94 | 100.00 ± 0.00 | 98.42 ± 1.63 | 9.09 ± 3.14 |
| PDC    | LinkerNet | 17.96± 0.33 | 100.00 ± 0.00 | 81.94 ± 3.97 | 11.11 ± 1.92 |
|        | DiffLinker | 20.33 ± 0.29 | 100.00 ± 0.00 | 28.56 ± 4.26 | 40.12 ± 5.66 |

**Linker Property and Quality Analysis.** Table 5 summarizes the evaluation results of linker properties and quality metrics.

**Linker quality:** Both models achieved high Pass 2D Filter rates on the PROTAC and ADC datasets, whereas performance on the PDC dataset was comparatively lower, reflecting the increased complexity of generating chemically valid linkers for these molecules.

**Synthetic accessibility:** We computed the synthetic accessibility (SA) scores for the generated PROTAC molecules; PDC and ADC datasets were excluded from this evaluation due to their conjugate nature. The two models obtained SA scores of 4.56 and 4.99, respectively, indicating moderate synthetic feasibility for the generated PROTAC linkers.

**Geometric and Chemical properties:** Using the ground truth as a reference, the similarity between generated molecules and reference molecules was assessed based on geometric and chemical properties, quantified by the Shape and Color RDKit similarity score ranging from 0 to 1. Higher scores indicate closer agreement with the reference. Both models achieved scores in the range of 0–0.34, suggesting that the generated molecules exhibit notable deviations from the reference molecules in chemical property space, highlighting areas for potential improvement in molecular fidelity. To further analyze the chemical similarity between the generated molecules and the reference molecules, we plotted the distribution histogram of the $SC_{RDKit}$ similarity scores for the generated molecules. Figure 6 indicates that the majority of generated molecules have $SC_{RDKit}$ scores in the range of 0–0.5. DiffLinker is capable of generating molecules with $SC_{RDKit}$ scores greater than 0.9 on both the PROTAC and PDC subsets, whereas LinkerNet fails to produce molecules with RDKit scores above 0.8 in any subset. Across all similarity thresholds, DiffLinker consistently outperforms LinkerNet.

**Table 5:** Chemical property-based results on three subsets in the HBDrug3D dataset. For each fragment pair on the test set, 100 candidates are generated for evaluation. Experiments were run for 3 times.

| Subset | Method | Pass 2D Filter, % | Average $SC_{RDKit}$ | SA |
|--------|--------|-------------------|----------------------|-----|
| PROTAC | LinkerNet | 93.77 ± 0.97 | 0.14 ± 0.01 | 4.99 ± 0.03 |
|        | DiffLinker | 98.68 ± 0.17 | 0.24 ± 0.01 | 4.56 ± 0.01 |
| ADC    | LinkerNet | 92.44 ± 0.65 | 0.34 ± 0.01 | –– |
|        | DiffLinker | 96.89 ± 3.84 | 0.31 ± 0.02 | –– |
| PDC    | LinkerNet | 80.60 ± 0.78 | 0.00 ± 0.00 | –– |
|        | DiffLinker | 66.03 ± 0.87 | 0.11 ± 0.01 | –– |

**Table 6:** $SC_{RDKit}$ score distributions of DiffLinker and LinkerNet on three subsets.

| $SC_{RDKit}$ | PROTAC | | ADC | | PDC | |
|---|---|---|---|---|---|---|
| | LinkerNet | DiffLinker | LinkerNet | DiffLinker | LinkerNet | DiffLinker |
| > 0.9 | 0.00% | 10.03% | 0.00% | 0.00% | 0.00% | 0.25% |
| > 0.8 | 0.00% | 16.18% | 0.00% | 0.00% | 0.00% | 0.97% |
| > 0.7 | 0.03% | 16.99% | 0.00% | 1.25% | 0.00% | 1.63% |
| > 0.6 | 1.53% | 18.49% | 1.63% | 1.87% | 0.06% | 5.11% |
| > 0.5 | 11.36% | 22.01% | 6.01% | 3.12% | 0.47% | 15.44% |
| > 0.4 | 35.91% | 40.88% | 25.62% | 14.02% | 3.64% | 33.21% |
| > 0.3 | 61.66% | 66.39% | 60.67% | 54.21% | 33.06% | 42.57% |
| > 0.2 | 69.43% | 72.92% | 97.30% | 96.26% | 54.68% | 47.33% |
| > 0.1 | 70.26% | 73.23% | 99.22% | 98.13% | 62.59% | 55.41% |

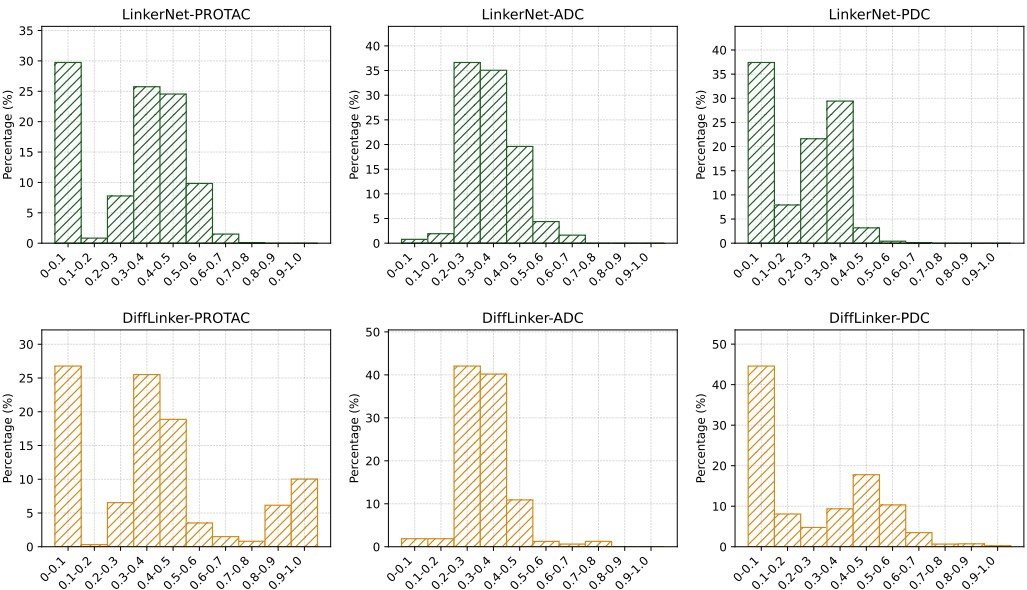

**Figure 6:** $SC_{RDKit}$ score distributions of DiffLinker and LinkerNet on three subsets.

## 6 CONCLUSION AND LIMITATION

Collectively, our analysis underscores the distinct complexity of heterobifunctional drug linker design compared to traditional small-molecule tasks, warranting its treatment as an independent research field. To support advancements in this domain, we developed HBDrug3D, a high-quality 3D dataset spanning PROTACs, ADCs, and PDCs, and established standardized protocols and evaluation metrics to enable rigorous and reproducible benchmarking. Our results highlight the pressing need for more robust and generalizable 3D structure-based models to bridge the methodological gap between small-molecule and heterobifunctional drug design.

Nonetheless, several limitations remain. First, expanding HBDrug3D to include additional promising modalities such as Aptamer-Drug Conjugates (ApDCs) is a key direction for future work. Second, due to the absence of wet-lab validation, current evaluation metrics based on computational approximations often fail to capture the nuanced physicochemical requirements of heterobifunctional linkers. Properties such as flexibility, hydrophilicity, and metabolic stability are challenging to assess with accuracy. Future efforts will focus on leveraging AI to develop more precise and property-aware evaluation strategies.

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

APPENDIX

## A  CONJUGATION SITE TYPE

This appendix catalogs the antibody-drug conjugate (ADC) linkage strategies analyzed in this study, with 17 conjugation sites shown in

- Conventional chemical conjugation (cysteine thiol-maleimide, lysine NHS ester)
- Enzymatic approaches (Sortase A transpeptidation, microbial transglutaminase amidation)
- Glycoengineering techniques (LacNAc-based remodeling)
- Genetic incorporation strategies (unnatural amino acids, SMARTag formylglycine)

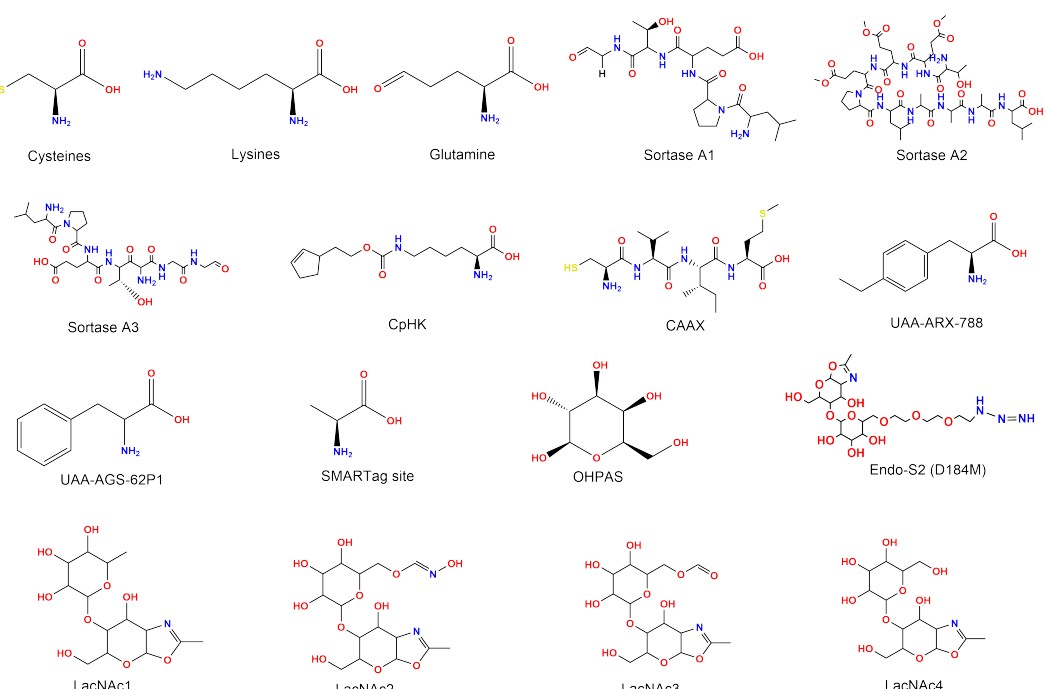

**Figure 7:** Chemical Structures of 17 Conjugation Sites in the ADCdb Dataset.

## B  ANTIBODY-DRUG CONJUGATE METHODS

**Table 7:** Antibody-Drug Conjugate Methods

| Conjugate Site | Conjugate Mechanism | Conjugate Strategies |
|---|---|---|
| **Cysteine** | Thiol-reactive chemistry (e.g. maleimide) with reduced -SH groups (Tsuchikama & An, 2018) | • Random (interchain disulfides)
• Engineered (THIOMAB™ (Junutula et al., 2008)/H38C2(Rader et al., 2003))
• Glycan-directed |

*Continued on next page*

| Conjugate Site | Conjugate Mechanism | Conjugate Strategies |
|---|---|---|
| | *Continued from previous page* | |
| **Lysine** | NHS ester amidation with $\epsilon$-amino groups | • Random (surface lysines)
• Engineered sites
• Glycan-proximal |
| **Glutamine (mTG)** | mTG-mediated transamidation at Q295 (native) or engineered Q-sites (Lhospice et al., 2015) | • Native Q295/Q311/Q438 (Li et al., 2023)
• Engineered (N→Q mutations)
• C-terminal tags (Wong et al., 2018; King et al., 2018) |
| **Sortase A** | LPXTG motif cleavage + oligoglycine (GGGY) ligation (Jäger et al., 2022) | • C-terminal (LPXTG-proteins)
• N-terminal (Gly$_3$-payloads) (Yap et al., 2019) |
| **CpHK** | A reactive diene for Diels-Alder (DA) reactions (Ting et al., 2022) | • Fast kinetics ($>10^4$ M$^{-1}$s$^{-1}$)
• Live-cell compatible |
| **CAAX** | Prenyltransferase adds azido-isoprenyl to CVIM motif (Min et al., 2020) | • Light chain C-terminal (Lim, 2020)
• Thioether click conjugation |
| **UAA** | Genetic incorporation of bioorthogonal handles (azide/alkyne) (Shiomi et al., 2021; Humphreys et al., 2015; VanBrunt et al., 2015; Rudra-Ganguly et al., 2016) | • Direct encoding (pAcF, AzF)
• Click chemistry conjugation |
| **SMARTag** | FGE-oxidized CxPxR → fGly aldehyde + HIPS ligation (Rabuka et al., 2012) | • Natural CxPxR tags
• Engineered insertions |
| **OHPAS** | One aryl acting as a payload, a self-immolative sulfate unit having a latent phenol function at the ortho position (Park et al., 2019) | • Tumor-activated release
• Plasma-stable design |
| **Endo-S2** | D184M mutant: transglycosylates azido-GlcNAc to Asn297 (Zhang et al., 2022) | • One-step remodeling
• Homogeneous DAR2 |
| **LacNAc** | Endo-S2 inserts azido-LacNAc oxazoline + click chemistry (Shi et al., 2022) | • Fc-specific
• 6-azidoGalNAc carrier |

## C  IMPLEMENTATION OF THE BASELINES

We outline the implementation of four baselines below:

**DeLinker** We utilized the original source code (https://github.com/oxpig/DeLinker) under TensorFlow 1.10 framework, maintaining the same framework and hyperparameter configurations as the published work.

**3DLinker** We utilized the original source code (https://github.com/YinanHuang/3DLinker) under PyTorch 1.11.0 framework, maintaining the same framework and hyperparameter configurations as the published work.

**DiffLinker** We implemented DiffLinker using the official source code (https://github.com/igashov/DiffLinker) with PyTorch 2.0, strictly maintaining the original model architecture and hyperparameters. The only modification involved adjusting the number of training epochs (from 500 to 600) to ensure proper convergence while keeping all other training parameters unchanged

(batch size=32, learning rate=1e-4, 1000 diffusion steps). The model was trained on NVIDIA A100 GPUs with mixed-precision acceleration, following the same data preprocessing pipeline as described in the original work.

**LinkerNet** We faithfully reproduced LinkerNet using the original source code (`https://github.com/guanjq/LinkerNet`) implemented in PyTorch 2.5.1, maintaining all published model architectures without modification.

All experiments were carried out using the default configuration (500 diffusion steps, initialized with the provided checkpoints pre-trained on ZINC), executed on NVIDIA RTX 4090 GPUs with full precision floating-point operations with modifications for different training scenarios:

- For PROTAC training: Batch size set to 4, learning rate of 5e-4 (with cosine decay), initialized with checkpoints pre-trained on ZINC
- For PDC training: Batch size of 4, learning rate of 5e-4 (with cosine decay)
- For ADC training: Batch size reduced to 2, learning rate adjusted to 1e-4 (with cosine decay)

# D  THE USE OF LARGE LANGUAGE MODELS

We employ large language models exclusively for language editing, which is limited to polishing text to improve readability. No language models contributed to the development of research ideas, analysis, models, or interpretation of results.

