# OpenReview forum: "HBDrug3D: A Dataset and Benchmark for AI-Driven Heterobifunctional Molecule Design"
_ICLR.cc/2026/Conference — Submitted to ICLR 2026_

### Official Review · Reviewer_veoK · 2025-10-30

**Soundness:** 3
**Presentation:** 4
**Contribution:** 2
**Rating:** 4
**Confidence:** 4

**Summary:**

The paper introduces HBDrug3D, a benchmark dataset for AI-driven heterobifunctional molecule design, covering PROTACs, ADCs, and PDCs. It provides a unified molecular representation, standardized conformational sampling, and consistent evaluation metrics for linker generation. The authors benchmark mainly two state-of-the-art generative models and analyze their performance, revealing the challenges of modeling complex linker geometries and cross-modality generalization.

**Strengths:**

-  Presents a carefully curated 3D dataset with standardized conformations

- The paper is clearly written and easy to follow, with well-designed figures, making it accessible to cross-discipline readers.

-  Addresses an important and timely problem in modern drug discovery, providing a foundation for future AI-driven linker design research.

**Weaknesses:**

1. The "Evaluation Metrics" section is overly brief and lacks formal definitions. It is unclear, for example, whether validity is determined by RDKit sanitization and valence checks and how novelty and uniqueness handle canonicalization, stereochemistry, or tautomers.

2. Although the paper presents four baselines (DeLinker, 3DLinker, DiffLinker, and LinkerNet), only DiffLinker and LinkerNet are successfully evaluated. DeLinker and 3DLinker are reported to have “failed to generate valid linkers”, yet no quantitative evidence or adaptation attempts are provided. The absence of a failure analysis makes it unclear whether the problem arises from dataset complexity, model capacity, or implementation details.

3. While the authors emphasize a unified heterobifunctional framework, all experiments are conducted separately for the PROTAC, ADC, and PDC subsets. No multi-task or cross-modality training is explored. As a result, the “unified” aspect appears to apply only to data formatting rather than to modeling or analysis.

**Questions:**

1. Although PROTACs, ADCs, and PDCs can all be categorized as heterobifunctional molecules, their linker design objectives differ substantially. For instance, PROTAC linkers must maintain a stable ternary complex geometry, whereas ADC linkers require controlled cleavage to release the payload. What is the biological or modeling justification for combining them into one unified dataset?

2. How is the scaffold defined in the “scaffold-based strategy” (Line 320)? Which cheminformatics package and normalization procedure were used? Is the scaffold split applied to the entire heterobifunctional molecule or only to the linker portion? Given the large size of ADCs, is this strategy appropriate, and how is potential data leakage avoided when the same linker occurs with different fragments?

3. Since the baseline models require 3D conformations as input, they must rely on pre-existing datasets or internal conformation-generation pipelines. Could the authors briefly summarize the sources, dataset sizes, and generation methods used in previous works (DeLinker, 3DLinker, DiffLinker, and LinkerNet etc.) and explain how these differ from the HBDrug3D protocol?

---

### Official Review · Reviewer_1upa · 2025-11-01

**Soundness:** 2
**Presentation:** 3
**Contribution:** 2
**Rating:** 4
**Confidence:** 3

**Summary:**

This paper introduces HBDrug3D, a new benchmark dataset for designing linkers in heterobifunctional drugs like PROTACs, ADCs, and PDCs. The authors argue that designing linkers for these molecules is more complex than for traditional small molecules. To create HBDrug3D, they collected data from existing public databases (PROTAC-DB, ADCdb, and PDCdb), standardized the formats, and generated a large number of high-quality 3D conformations using the Schrödinger OPLS4 force field. The final dataset includes 6,279 molecules and 59,314 conformations. The paper also provides a standardized evaluation framework and benchmarks several state-of-the-art linker design models, revealing that current methods have significant room for improvement, especially in generating diverse and accurate 3D structures.

**Strengths:**

1. The authors have created a valuable resource by aggregating, curating, and standardizing data from multiple sources into a unified, high-quality 3D conformational dataset. This effort helps to establish a much-needed benchmark for the field.

2. The inclusion of a standardized codebase for data processing, training, and evaluation is a significant contribution that will lower the barrier to entry for other researchers and promote reproducible research.

3. The paper is well-structured and clearly written, effectively motivating the need for the HBDrug3D dataset and benchmark.

**Weaknesses:**

1. **No Raw Data:** The primary contribution of HBDrug3D is the generation of 3D conformations and the unification of existing data, rather than the collection of new experimental data. The raw molecular structures are sourced from established public databases.

2. **Lack of Comparative Benchmarking:** The evaluation demonstrates that existing models struggle with the HBDrug3D dataset. However, all models were retrained exclusively on HBDrug3D. A more compelling experiment would be to compare the performance of models trained on HBDrug3D against the same models trained on more general small-molecule datasets like ZINC. This would more directly validate the necessity and benefit of a specialized dataset for designing heterobifunctional molecules.

3. **Absence of Ternary Complex Structures:** The dataset is composed of isolated molecules (PROTACs, ADCs, PDCs). However, the function of these molecules, such as PROTACs, is critically dependent on the formation of a ternary complex (e.g., Target Protein-PROTAC-E3 Ligase). The conformation of the linker is heavily influenced by the protein-protein interface it is mediating. By providing only the molecule's structure in isolation, the dataset omits the most critical biological context for rational linker design. Notably, databases like PROTAC-DB, which was a source for this work, do contain an increasing number of experimentally determined and predicted ternary complex structures.

**Questions:**

1. A quantitative analysis of the accuracy of the generated conformations would strengthen the paper. While the OPLS4 force field is a standard method, it would be beneficial to compare the generated low-energy conformations against any available experimentally determined structures (e.g., from the PDB) for a subset of the molecules to provide a measure of how well the computational method recapitulates reality. This would give users of the dataset more confidence in the quality of the 3D data.

---

### Official Review · Reviewer_TSKn · 2025-11-01

**Soundness:** 2
**Presentation:** 2
**Contribution:** 2
**Rating:** 2
**Confidence:** 3

**Summary:**

This paper introduces HBDrug3D, a comprehensive 3D dataset and benchmarking platform for AI-driven heterobifunctional drug linker design. It unifies data from PROTACs, ADCs, and PDCs, providing 6,279 molecules and 59,314 conformations generated and curated with standardized procedures. The benchmark defines consistent evaluation metrics for chemical validity, 3D similarity, and molecular properties, and systematically assesses several state-of-the-art generative models. Results show that current models perform poorly on complex linker structures, emphasizing the need for more generalizable 3D-aware generative approaches.

**Strengths:**

1. The paper formalizes heterobifunctional linker design as a distinct task and builds the first large-scale 3D dataset covering PROTACs, ADCs, and PDCs.
2. The authors harmonize chemical representations, apply OPLS4-based conformer sampling, and perform strict RMSD filtering, ensuring diverse and physically accurate 3D structures.
3. The study establishes consistent experimental protocols and evaluation metrics across modalities, enabling fair comparison and reproducible performance analysis of linker design models.

**Weaknesses:**

1. The paper benchmarks existing generative models (DeLinker, 3DLinker, DiffLinker, and LinkerNet) trained from scratch on HBDrug3D. Since these models were originally pretrained on large small-molecule datasets like ZINC, fine-tuning them on HBDrug3D could preserve their broader understanding of chemical space and reveal how transferable their learned geometric and chemical priors are to heterobifunctional molecules.

2. The comparison between DiffLinker and LinkerNet in Table 6 and Figure 6 lacks statistical rigor. The authors draw performance conclusions solely from visual and descriptive trends without conducting any statistical tests or confidence analyses to verify whether the observed differences in SCRDKit similarity are significant.

3. Table 6 is largely redundant because it reports the same SCRDKit similarity distributions already visualized in Figure 6. The figure conveys the performance gap between DiffLinker and LinkerNet more clearly, making the detailed numeric table unnecessary for the main text.

4. The paper does not clearly emphasize which evaluation metrics best represent model performance. Core metrics such as validity, recovery, and 3D similarity (SCRDKit) should probably be prioritized, as they directly measure whether generated linkers are chemically correct, structurally faithful, and geometrically realistic for real-world drug discovery. In contrast, uniqueness and novelty are less informative for this task, and their inclusion without clear analysis may distract readers from the main evaluation focus.

**Questions:**

1. Could the authors explain why the DeLinker and 3DLinker models fail to generate valid linkers, even when trained from scratch on valid molecules in HBDrug3D?

2. Why does the uniqueness of DiffLinker drop significantly when trained on the ADC subset?

3. What is the trade-off between uniqueness and recovery, specifically in the context of heterobifunctional drug linker design?

---

### Meta-Review · Area_Chair_dA6S · 2026-01-02

**Summary:**

HBDrug3D is a dataset designed to benchmark AI-driven molecule design of heterobifunctional molecules such as PROTACs, ADCs and PDCs. The paper addressed a significant gap in the benchmarks.

Reviewers had significant concerns. Authors have not provided Rebuttal. Given that there are significant concerns that are left unaddressed, I have to recommend rejection at this satge.

**Reviewer Concerns:**

Key concerns:

- Reviewer 1upa noted lack of biological context
- Reviewers TSKn and 1upa were concerns with limited novelty of the paper in that it combines existing datasets
- Reviewers TSKn and veoK asked for additional analysis of failure of the tested methods

**Reviewer Scores:**

No rebuttal was provided.

---

### Decision · Program_Chairs · 2026-01-26

Reject